# Molecular reconstruction of recurrent evolutionary switching in olfactory receptor specificity

**Lucia L Prieto-Godino[1,2]\*, Hayden R Schmidt[1], Richard Benton[1]\***

[1]Center for Integrative Genomics, Faculty of Biology and Medicine, University of Lausanne, Lausanne, Switzerland; [2]The Francis Crick Institute, London, United Kingdom

**Abstract** Olfactory receptor repertoires exhibit remarkable functional diversity, but how these proteins have evolved is poorly understood. Through analysis of extant and ancestrally reconstructed drosophilid olfactory receptors from the Ionotropic receptor (Ir) family, we investigated evolution of two organic acid-sensing receptors, Ir75a and Ir75b. Despite their low amino acid identity, we identify a common 'hotspot' in their ligand-binding pocket that has a major effect on changing the specificity of both Irs, as well as at least two distinct functional transitions in Ir75a during evolution. Moreover, we show that odor specificity is refined by changes in additional, receptor-specific sites, including those outside the ligand-binding pocket. Our work reveals how a core, common determinant of ligand-tuning acts within epistatic and allosteric networks of substitutions to lead to functional evolution of olfactory receptors.

## Editor's evaluation

This study investigates evolutionary changes in ligand preference that occur in an olfactory receptor (IR75a) across the *Drosophila* phylogeny. The authors find that IR75a displays different odor preferences, for acetic acid or butyric acid, across *Drosophila* species, and link odor preference to particular protein mutations in the receptor. Reconstruction of a putative ancestral IR75a revises the timeline for IR75a evolution, and structural modeling suggests how mutations alter odor preference.

## Introduction

Among the senses, olfaction is particularly flexible over evolutionary time, enabling animals to adapt their recognition of the vast, ever-changing universe of volatile chemicals in the environment (*Bargmann, 2006*; *Ramdya and Benton, 2010*). This flexibility is reflected in the evolution of large, divergent families of olfactory receptors with different odor tuning properties. Several functional surveys of receptor repertoires in vertebrates (e.g., humans and the house mouse *Mus musculus*) and insects (e.g., the vinegar fly *Drosophila melanogaster* and the malaria mosquito *Anopheles gambiae*) have identified ligands for many receptors (*Carey et al., 2010*; *Hallem and Carlson, 2006*; *Saito et al., 2009*). Moreover, comparative sequence and functional analyses of orthologous receptors across species have started to identify amino acid differences that can explain species-specific receptor tuning properties (*Adipietro et al., 2012*; *Auer et al., 2020*; *Del Mármol et al., 2021*; *Leary et al., 2012*; *Mainland et al., 2014*; *Prieto-Godino et al., 2017*; *Yang et al., 2017*). However, the molecular basis of functional changes in receptors over evolutionary timescales – and whether common principles in this process exist between different receptors – remains unclear.

**\*For correspondence:**
lucia.prietogodino@crick.ac.uk (LLP-G);
Richard.Benton@unil.ch (RB)

**Competing interest:** The authors declare that no competing interests exist.

A powerful model to study olfactory receptor evolution is the Ionotropic receptor (Ir) repertoire, a protostomian chemosensory subfamily of ionotropic glutamate receptors (iGluRs) (*Benton et al., 2009*; *Croset et al., 2010*; *Ni, 2020*; *Rytz et al., 2013*). Although Irs and iGluRs have limited amino acid sequence identity, their overall conserved (predicted) secondary and tertiary structural organization suggests that the chemosensory receptors share many mechanistic similarities with their iGluR ancestors (*Abuin et al., 2011*). The best-characterized Irs are predicted to be heterotetramers formed of two subunits of a conserved coreceptor and two subunits of a 'tuning' receptor (*Abuin et al., 2011*; *Abuin et al., 2019*). The latter are more variable in sequence both within and between species, particularly in the extracellular ligand-binding domain (LBD), consistent with their diverse odor recognition properties.

Functional characterization of Irs in different drosophilid species has revealed orthologous receptors that have distinct odor recognition properties (*Prieto-Godino et al., 2017*; *Prieto-Godino et al., 2016*), notably Ir75a, a receptor that probably originated in the Neodipteran ancestor (>200 million years ago) and its paralog Ir75b, which arose through duplication of *Ir75a* in the Drosophilidae ancestor (~60–70 million years ago) (*Croset et al., 2010*; *Prieto-Godino et al., 2017*). In *D. melanogaster* and *D. simulans*, two cosmopolitan species that feed on a wide range of fermented fruit, these receptors exhibit different sensitivity toward carboxylic acids: Ir75a is tuned predominantly to acetic acid, while

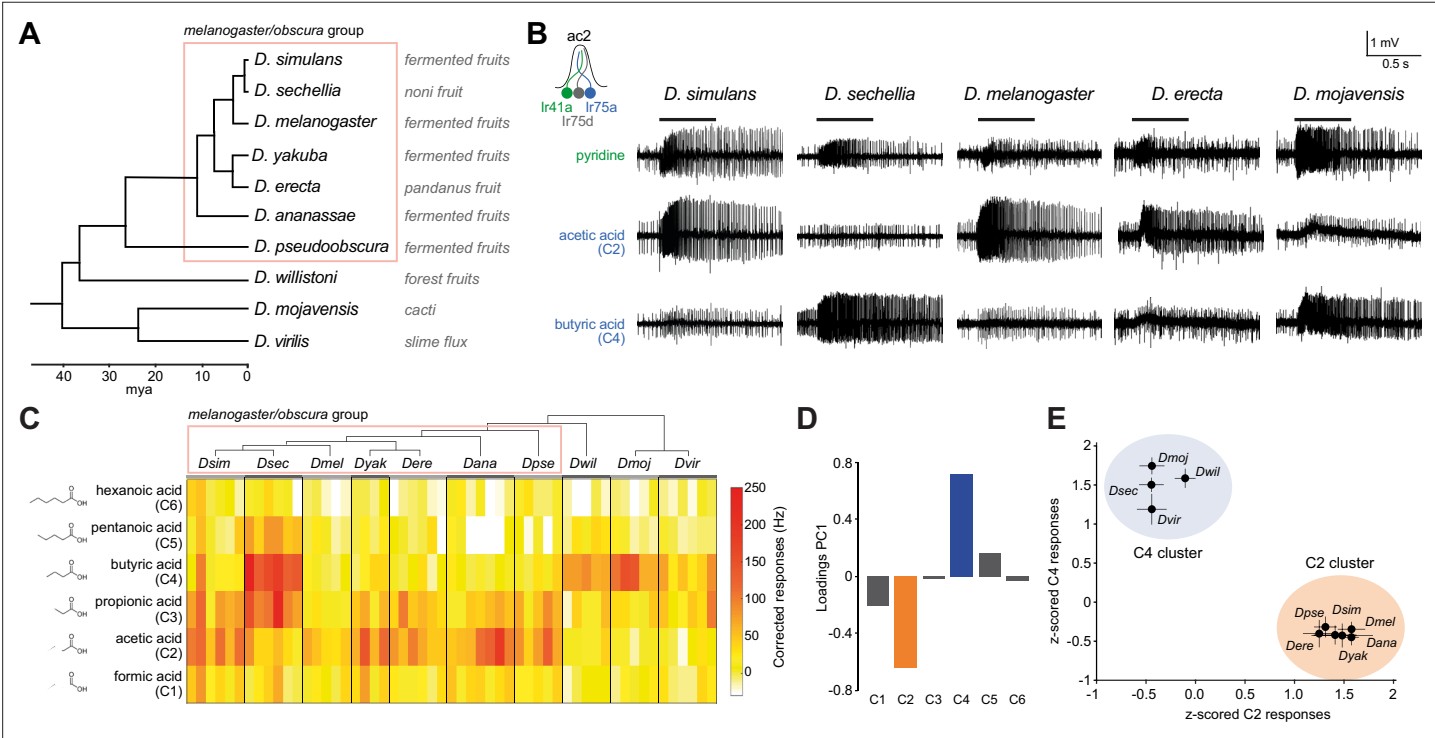

**Figure 1.** Evolution of olfactory responses of Ir75a neurons across the drosophilid phylogeny. (**A**) Phylogeny of analyzed drosophilid species and their known ecological niches adapted from *Figure 2*, *Markow, 2015*. (**B**) Representative traces of extracellular recordings of neuronal responses in antennal coeloconic 2 (ac2) sensilla (schematized top left) to the indicated odors in different drosophilid species. ac2 sensilla were identified based on their morphology, their location on the antennal surface – either near the entry to the sacculus or in the distal posterior part intermingled with ac3 sensilla (*Silbering et al., 2011*) – their pattern of basal firing, and the conserved responses of the Ir41a neuron toward pyridine and the lack of response to octanol which is detected by the Or35a neuron in ac3 sensilla (*Yao et al., 2005*). (**C**) Heatmap of electrophysiological solvent-corrected responses (see Materials and methods) of ac2 sensilla of all species in (**A**) to a series of linear carboxylic acids; the color scale is on the right. Each rectangle represents the responses measured in a single sensillum. (**D**) Loadings of the first principal component of all responses shown in (**C**). (**E**) Responses of ac2 sensilla for all species plotted as *z*-scored responses (see Materials and methods) of C2 against *z*-scored responses to C4. This plotting reveals two clear clusters with species responding maximally to C2 (orange) or C4 (blue).

Figure 1A is adapted. (A) Phylogeny of analyzed drosophilid species and their known ecological niches (adapted from Figure 2, Markow, 2015)..

The online version of this article includes the following figure supplement(s) for figure 1:

**Source data 1.** Data for *Figure 1* showing the solvent-corrected spikes/s responses of antennal coeloconic 2 (ac2) sensilla for each of the species (*Figure 1C*).

Ir75b responds maximally to butyric acid (*Prieto-Godino et al., 2016*; *Prieto-Godino et al., 2017*; *Silbering et al., 2011*). By contrast, in the closely related island endemic *D. sechellia*, which feeds and breeds exclusively on the ripe noni fruit of the *Morinda citrifolia* shrub (*Figure 1A*), Ir75a and Ir75b preferentially respond to butyric acid and hexanoic acid, respectively (*Prieto-Godino et al., 2017*; *Prieto-Godino et al., 2016*). These differences are likely to be ecologically significant: for example, acetic acid is a key product of microbial fermentation of vegetal material and regulates numerous behaviors in *D. melanogaster* including attraction (*Becher et al., 2010*), sexual receptivity (*Gorter et al., 2016*), and oviposition (*Joseph et al., 2009*; *Kim et al., 2018*). Hexanoic acid is a dominant component of noni fruit and elicits attractive behaviors in *D. sechellia* (*Amlou et al., 1998*; *Dekker et al., 2006*; *Prieto-Godino et al., 2017*). In this work, we combine comparative in vivo functional analyses of these receptors across the drosophilid phylogeny, with ancestral sequence reconstruction, site-directed mutagenesis and protein modeling to investigate their evolution.

## Results

### Evolution of olfactory responses of Ir75a across the drosophilid phylogeny

The distinct responses of *D. sechellia* Ir75a (*Dsec*Ir75a) compared to orthologs in its two generalist cousins (*Prieto-Godino et al., 2016*) suggested that acetic acid sensing was the ancestral function of Ir75a. We tested this hypothesis by measuring odor-evoked responses of Ir75a-expressing olfactory sensory neurons (OSNs) across the drosophilid phylogeny, representing >40 million years divergence time (*Figure 1A–C*, see Materials and methods). As stimuli we used a panel of linear carboxylic acids spanning from one- to six-carbon chains (hereafter abbreviated to C1–C6, where acetic acid is C2 and butyric acid is C4). All tested species within the *melanogaster/obscura* group (except for *D. sechellia*) displayed strongest responses to C2, similar to *D. melanogaster* and *D. simulans* (*Figure 1B, C*). Unexpectedly, the responses of more divergent species were similar to those of *D. sechellia*, exhibiting strongest responses to C4 (*Figure 1B, C*).

To simplify data visualization and discern in an unbiased way which odors contribute maximally to differential tuning of Ir75a neurons across species, we performed principal component analysis (PCA) on their response profiles. The first principal component (PC1) explains 67.5 % of the variance in the data, mostly capturing the inverse variation between C2 and C4 (*Figure 1D*). When plotting the C2 and C4 responses against each other, Ir75a neurons of different species segregated into two clusters with either high responses to C2 and low to C4 or vice versa (*Figure 1E*). The clustering matched well the phylogeny with the exception of *D. sechellia*, which grouped together with *D. willistoni*, *D. mojavensis*, and *D. virilis* (*Figure 1A, E*). These observations suggested a new model in which the ancestral drosophilid Ir75a was predominantly a C4 sensor that evolved to become a C2 sensor in the last common ancestor of the *melanogaster* and *obscura* groups, before reverting to an ancestral-like state in *D. sechellia* (*Figure 2A*).

### Functional analysis of ancestrally reconstructed Ir75a

To test this hypothesis, we 'resurrected' the ancestral Ir75a receptors at the ancestral drosophilid and *melanogaster/obscura* group phylogenetic nodes by inferring their sequence via maximum likelihood (*Randall et al., 2016*) from the sequence of orthologs from 16 extant species (*Figure 2A* and *Figure 2—figure supplement 1* and Materials and methods). We synthesized genes encoding the inferred ancestral proteins – termed here Ir75a[Dros] and Ir75a[mel-obs], respectively – and integrated these into a common genomic location to avoid differential positional influence on their expression. These transgenes were expressed individually in the *D. melanogaster* 'Ir decoder neuron,' an OSN that lacks the endogenous tuning receptor subunit but expresses the Ir8a coreceptor (*Abuin et al., 2011*; *Grosjean et al., 2011*; *Prieto-Godino et al., 2016*). When expressed in this system, the tuning profiles of *D. melanogaster* Ir75a (*Dmel*Ir75a) and *Dsec*Ir75a recapitulate those of the receptors expressed in their endogenous neurons, with strongest responses to C2 and C4, respectively (*Figure 2B–C*), consistent with previous observations (*Prieto-Godino et al., 2016*). As predicted, Ir75a[mel-obs] responded similarly to *Dmel*Ir75a (*Figure 2B, C*). Importantly, the more ancient, resurrected receptor, Ir75a[Dros], had a tuning profile that was almost identical to that of *Dsec*Ir75a, with maximal responses to C4

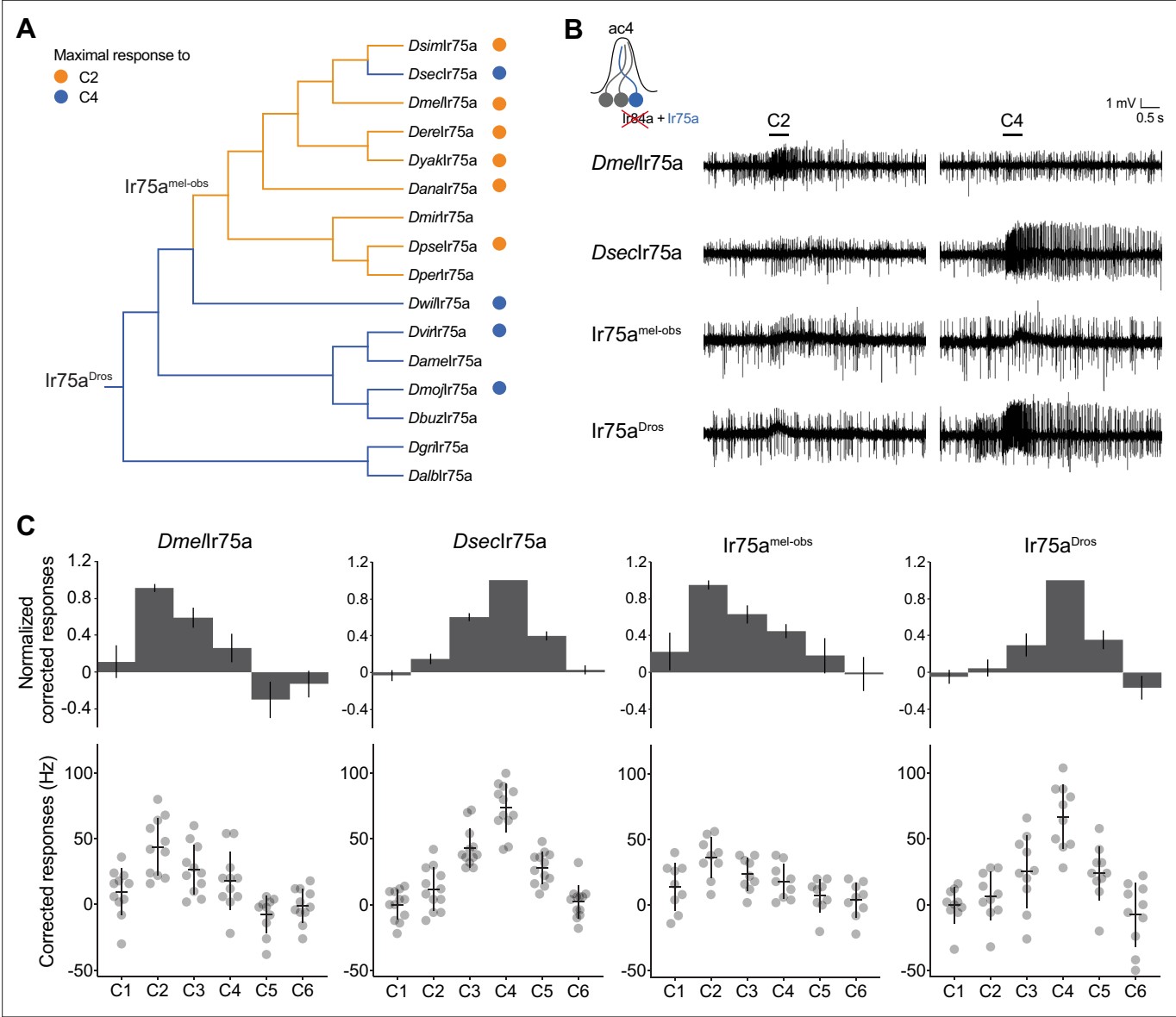

**Figure 2.** Ancestral sequence reconstruction of Ir75a. (**A**) Phylogeny of the protein sequences of drosophilid Ir75a orthologs used for the reconstruction of ancestral nodes (Ir75a^Dros and Ir75a^mel-obs) (see **Figure 2—figure supplement 1**). The orange and blue dots indicate maximal responses of the corresponding neurons to C2 and C4, respectively (**Figure 1C**). The branches of the tree have been similarly color-coded according to predicted receptor responses based on parsimony. *Drosophila* species abbreviations (where not presented in **Figure 1**): Dmir (*miranda*), Dper (*persimilis*), Dame (*americana*), Dbuz (*buzzattii*), Dgri (*grimshawi*), and Dalb (*albomicans*). (**B**) Representative traces of extracellular recordings of neuronal responses to the indicated odors of receptors expressed in the Ir decoder neuron (see text). Genotypes are of the form: *UAS-xxx/UAS-xxx;Ir84a^Gal4/Ir84a^Gal4*, here and in all subsequent figures. (**C**) Quantification of responses of the indicated receptors expressed in the Ir decoder neuron. In this and subsequent similar figure panels, the top row shows barplots of responses normalized by maximal response, and the bottom row shows individual datapoints, mean and standard error of mean (SEM) of raw solvent-corrected responses to odor stimuli.

The online version of this article includes the following figure supplement(s) for figure 2:

**Source data 1.** Data for **Figure 2C**, responses in spikes/s of antennal coeloconic 2 (ac2) sensilla from each of the genotypes, and normalized with respect to the maximal response of each sensilla.

**Figure supplement 1.** Alignment of Ir75a orthologs.

(*Figure 2B, C*). These results indicate that Ir75a has switched responsiveness at least twice during its evolutionary history, from principally C4-sensing to C2-sensing and, in *D. sechellia*, back again.

## Identifying the molecular basis of the functional evolution of Ir75a

To determine the molecular basis of the evolution of Ir75a responses, we aligned the sequences of Ir75a$^{mel-obs}$ and Ir75a$^{Dros}$. 114 positions in these proteins exhibit different amino acids (82 % identity), of which 45 are located within the LBD (*Figure 2—figure supplement 1*). As this level of divergence precluded straightforward experimental determination of the relevant sites, we first approached the problem by focusing on the more recent 'reverse' transition of C2-sensing to C4-sensing that occurred on the branch leading to *Dsec*Ir75a. We had previously narrowed down this tuning switch to three amino acid positions within the internal pocket of the bilobed (S1–S2) LBD (*Prieto-Godino et al., 2016*; *Figure 3A*). Simultaneous substitution of these sites in *Dmel*Ir75a with the residues found in *Dsec*Ir75a (i.e., T289S, Q536K, and F538L) produced a receptor that faithfully recapitulated the response properties of *Dsec*Ir75a when assessed in the Ir decoder neuron (*Figure 3B*), as previously described (*Prieto-Godino et al., 2016*). Furthermore, reverse amino acid substitutions in *Dsec*Ir75a (S289T, K536Q, and L538F) conferred response properties characteristic of *Dmel*Ir75a (*Figure 3B*). These results indicate that the change in tuning is encoded entirely within these three sites. Thus, there are only $2^3 = 8$ variants to transit between *Dmel*Ir75a and *Dsec*Ir75a (including the wild-type sequences) – compared to $2^{114} = 2 \times 10^{34}$ possible variants between Ir75a$^{Dros}$ and Ir75a$^{mel-obs}$ – offering an excellent opportunity to study the functional evolution of an olfactory receptor.

We generated versions of *Dmel*Ir75a in which each of these three sites was substituted individually as well the three possible double substitutions. All single amino acid changes had an impact on receptor responses, shifting tuning toward C4 (and C3) to different extents, while still retaining some sensitivity to C2 (*Figure 3C*). Double substitutions showed further shifts toward the *Dsec*Ir75a tuning profiles but to varying degrees: *Dmel*Ir75a$^{T289S,Q536K}$ is a broadly tuned receptor that responds maximally (albeit weakly) to C2, C3, and C4, *Dmel*Ir75a$^{Q536K,F538L}$ displays maximum responses to both C3 and C4, and *Dmel*Ir75a$^{T289S,F538L}$ responds maximally to C4 alone (*Figure 3C*). To move beyond simple descriptions of the effects of these mutations, we used visualizations and analyses that give insights into the evolutionary landscape of this receptor, as described below.

## Mapping the evolutionary landscape of Ir75a

Evolutionary landscapes map genotypes onto a quantitative phenotype to illustrate evolutionary change as a 'navigation' process. The roughness of the landscape determines how accessible each of all possible paths are, and therefore the likelihood that evolution proceeds through each path (*Aguilar-Rodríguez et al., 2017*; *Wright, 1932*). We reasoned this approach was useful to understand the evolution of Ir75a as the lack of informative intraspecific genetic diversity within these receptor genes in drosophilid species (data not shown; see Materials and methods) precluded the direct identification of evolutionary intermediates.

Response properties of a receptor are multidimensional, where each odor defines a dimension. However, PCA of the responses of all *Dmel*Ir75a receptor mutants to all odors revealed that PC1 explains most of the variance (54.5%) and captures the inverse variation between C2 and C4 – similar to the Ir75a neuron response PCA across the drosophilid phylogeny (*Figure 1D*) – and, to a lesser extent, the covariation of C5 with C4 (*Figure 3D*). We therefore plotted the phenotypic data for each single, double, and triple *Dmel*Ir75a mutant in the C2 vs C4 space alone (*Figure 3E*) and in the PC1 space (*Figure 3F*), acknowledging these simplifications represent only part of the functional changes (albeit potentially the most important).

The C2 vs C4 plot enables visualization of all possible evolutionary paths by joining the stepwise mutations from *Dmel*Ir75a to *Dmel*Ir75a$^{T289S,Q536K,F538L}$. Most receptor variants are plotted close to the straight line (i.e., the shortest path) that joins the initial and final state, with the *Dmel*Ir75a$^{F538L}$ single mutant showing the largest individual 'step' along this line (*Figure 3E*). If we assume that all steps along this line would be favored by selection, this observation suggests the hypothesis that the F538L change might have been the first one of the three substitutions to occur, as de novo mutations with larger effect are typically substituted first followed by those with smaller effect (*Holder and Bull, 2001*; *Orr, 2005*). An alternative hypothesis is that T289S variant already existed, perhaps as standing variation in the population (despite the lack of evidence from sequences of extant populations): T289S

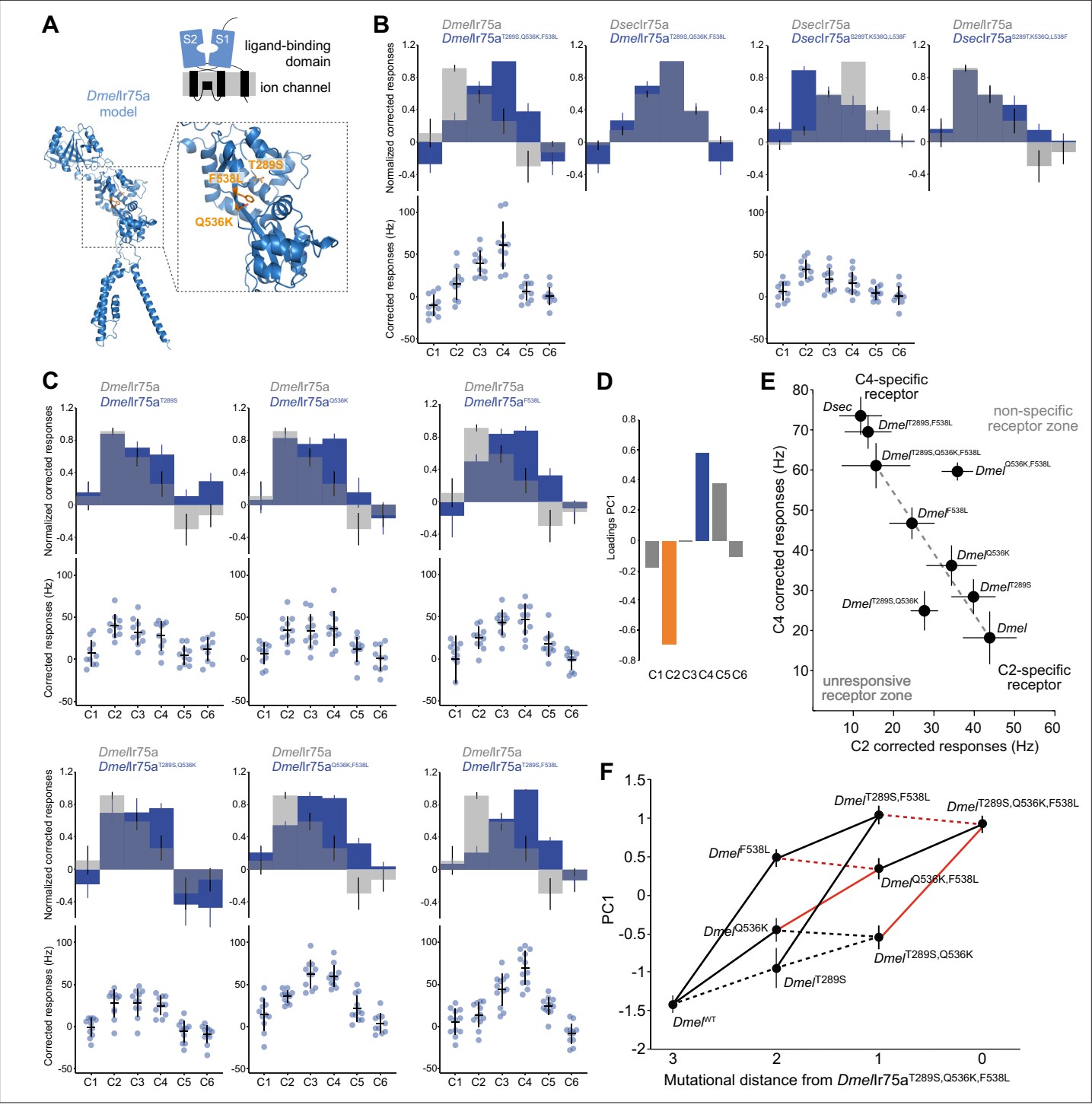

**Figure 3.** Mapping the evolutionary landscape of Ir75a. (**A**) Top: cartoon of the domain organization of Ir75a; bottom: protein model of *Dmel*Ir75a, generated by ab initio protein folding by AlphaFold (*Jumper et al., 2021*). A magnified view of the ligand-binding domain (LBD) is shown on the right, in which the three functionally important amino acid positions that differ between *Dmel*Ir75a and *Dsec*Ir75a are highlighted in orange. (**B, C**) Quantification of responses of the indicated receptor versions expressed in the Ir decoder neuron. (**D**) Loadings of the first principal component of all responses shown in (**B, C**). (**E**) Responses of each of the indicated receptor versions plotted in the C2 vs C4 space. (**F**) Visualization of epistasis and accessible mutational pathways. Each of the individual, double, and triple mutations are plotted in the first principal component (PC1) axis (error bars are standard error of the mean). Possible evolutionary paths join these mutants with lines. Solid lines indicate when the path joins two points that significantly increase PC1 value (i.e., increased responses to C4 and C5 and decreased responses to C2); dashed lines denote paths that, while accessible, do not lead to significantly increased PC1 values (see Source data, for statistical values). Red lines indicate the cases where two mutations

*Figure 3 continued on next page*

Figure 3 continued

interact epistatically when combined, that is, the combination of the two mutations is not equal to the expected response if their effects added linearly (see Materials and methods for details and source data for statistical values).

The online version of this article includes the following figure supplement(s) for figure 3:

**Source data 1.** Source data for *Figure 3B,C,F*.

---

has little phenotypic consequence by itself (*Figure 3C and E*), but it greatly augments the effect of the F538L substitution (*Figure 3C and E*). In this case, T289S and F538L would have reached fixation together in the same genetic background because their combined effect allows for a large adaptive leap from one peak (C2) to the other (C4).The Q536K change alone has an intermediate phenotype, but we suspect is unlikely to have been the first to occur, as combination with either other amino acid change leads to a less specific receptor (*Dmel*Ir75a$^{Q536K,F538L}$, lying above-right of the line), or a receptor with overall weak sensitivity (*Dmel*Ir75a$^{T289S,Q536K}$, lying below-left of the line) (*Figure 3C and E*).

To formalize this analysis, we calculated epistasis (i.e., nonadditive effects of different mutations), which can constrain the available evolutionary paths (*Aguilar-Rodríguez et al., 2017*; *Phillips, 2008*). Using the projection of our physiological responses onto PC1 as our phenotypic readout, we calculated for each of the combinations of mutants (i.e., single with single, or single with double) whether the sum of the effects of each of the starting substitutions on the wild-type background was significantly different from the effect of introducing both mutations simultaneously (see Materials and methods). Notably, negative magnitude epistasis was observed in all combinations of Q536K and F538L (whether T289S was also present the background or not) (*Figure 3F*, red lines). This observation supports our previous hypothesis that whether F538L or T289S was the first mutation to occur, the second one would be the other of these two, because Q536K incurs in negative epistasis when combined with F538L and it does not lead to a monotonic increase in responses when combined with T289S (*Figure 3F*). Furthermore, this visualization highlights the significant effect on tuning of the F538L substitution, as well as the minimal (statistically nonsignificant) effect of the T289S substitution, supporting its possible segregation in the population without deleterious effects. It remains unclear what, if any, is the function of Q536K; it is possible it was fixed by drift or affects some other aspect of receptor activity not analyzed here.

## A hotspot for evolution of Ir odor tuning

We next asked whether the knowledge of the molecular basis of the C2 → C4 tuning change on the *D. sechellia* branch offers insight into the ancestral switch from C4 to C2 in the last common ancestor of the *melanogaster/obscura* group. Strikingly, examination of the identity of residues aligned with *Dmel*Ir75a F538 in orthologs across the drosophilid phylogeny revealed a perfect correspondence between the identity of this position and the best agonist for the receptor: all Ir75a orthologs of species responding most strongly to C2 have an F, while those that respond to C4, have an L (*Figure 4A* and *Figure 2—figure supplement 1*). Such correspondence was not seen for amino acid identities at position 289 and 536: for example, *Dwil*Ir75a has a Q at the position equivalent to Q536 of *Dmel*Ir75a (*Figure 2—figure supplement 1*), but this species' Ir75a neurons respond to C4, like *D. sechellia* (*Figure 1C*).

These observations suggest that position 538 (or equivalent in orthologous sequences) has been a 'hotspot' for odor response evolution, changing from L to F in the *melanogaster/obscura* ancestor and then changing back on the *D. sechellia* branch. The reversion of amino acid identity in *Dsec*Ir75a is not due to an inverse mutation in the corresponding DNA sequence: a C → T mutation in codon position one led to the L to F substitution in the *melanogaster/obscura* ancestor, while in *D. sechellia* a C → A mutation in codon position three led, convergently, to restoration of the L-encoding codon (*Figure 4A*).

The important contribution of a single amino acid in Ir75a in determining the specificity for shorter- or longer-chain acids was reminiscent of our observation of the evolution of the paralogous receptor Ir75b: the difference in tuning of *Dmel*Ir75b and *Dsec*Ir75b to C4 and C6, respectively, is determined in large part by a T523S substitution in the LBD (*Prieto-Godino et al., 2017*). Ir75b and Ir75a exhibit only 38 % amino acid identity. However, alignment of these receptors revealed that position 523 in *Dmel*Ir75b corresponds precisely to the 538 hotspot in *Dmel*Ir75a (*Figure 4B*). Thus, evolution of

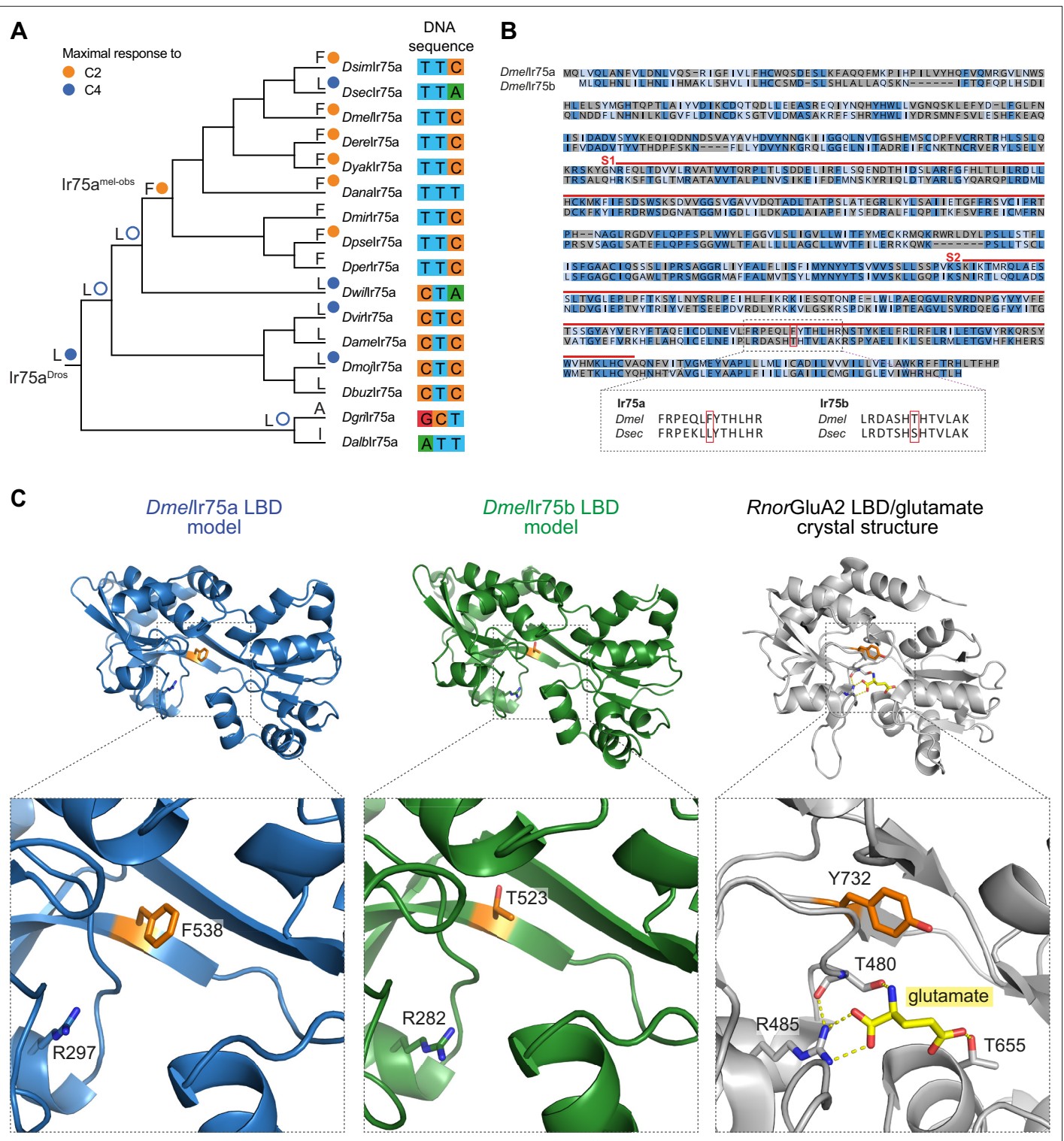

**Figure 4.** A hotspot for tuning Ionotropic receptor (Ir) sensitivity. (**A**) Phylogeny of species' Ir75a receptors used for the reconstruction of ancestral nodes. The inferred identity of hotspot mutation for key nodes is shown as well as for each of the extant receptors at the end of each leaf, together with the measured (solid circle) or predicted (empty circle) sensitivity. On the right, the nucleotides encoding the hotspot amino acid position in Ir75a across species. (**B**) Top: protein sequence alignment of *Dmel*Ir75a and *Dmel*Ir75b. Dark blue indicates identical residues, light blue indicates similar residues, and gray indicates dissimilar residues. The red lines indicate the spans of S1 and S2 lobes of the ligand-binding domain (LBD). The red box indicates the common amino acid position ('hotspot'). Bottom: separate alignments of the hotspot region in Ir75a and Ir75b for *D. melanogaster* and *D. sechellia* proteins showing that while the residue position is conserved, the identity of the amino acids is different for these two receptors. (**C**) Protein models of

*Figure 4 continued*

the *Dmel*Ir75a LBD and *Dmel*Ir75b LBD, and X-ray crystal structure of the *Rattus norvegicus* (*Rnor*) iGluR GluA2 LBD in complex with glutamate (PDB accession 1FTJ). The hotspot residue in the Irs and the equivalent residue (Y732) in *Rnor*GluA2 are shown in orange. A conserved arginine (R) residue that, in iGluRs, interacts with the glutamate ligand (yellow) is shown (see Discussion). Amino acid numbering corresponds to that of the full-length sequence.

novel specificities in two different receptors – which diverged from a common ancestor >60 million years ago (*Prieto-Godino et al., 2017*) – is specified by changes in the same site within their LBDs.

We mapped the position of these sites onto protein models of the *Dmel*Ir75a and *Dmel*Ir75b LBDs (*Figure 4C*). The hotspot residue projects into the cavity where odors are assumed to interact with Irs. However, it does not correspond with any known ligand-binding residue in iGluRs (see Discussion).

## Functional interactions between the Ir75b hotspot and surface residues of the LBD

Although mutation of the hotspot in *Dmel*Ir75b is sufficient to confer novel responses to C5 and C6, this single change did not fully recapitulate the response profile of *Dsec*Ir75b, as *Dmel*Ir75b$^{T523S}$ retained sensitivity to shorter chain acids (*Figure 5A*). Compared to *Dmel*Ir75b and *D. simulans* Ir75b, *Dsec*Ir75b contains three additional changes within the LBD pocket region (P473S, G492S, and A520T; like the hotspot, all are within the S2 lobe), but in our previous work none of these appeared to contribute substantially – individually or together – to the refinement of specificity for C6 (*Prieto-Godino et al., 2017*). The *Dsec*Ir75b LBD contains six additional derived residues located in S1 (*Prieto-Godino et al., 2017*), leading us to test whether these sites contribute to the observed changes in odor responses.

Replacement of the entire S1 lobe in *Dmel*Ir75b with that of *Dsec*Ir75b generated a receptor, *Dmel*Ir75b$^{DsecS1}$, with significantly increased responses to C6 and lower (albeit not statistically significant; *Figure 5—source data 1*") sensitivity to C3, when compared to *Dmel*Ir75b (*Figure 5A*). Addition of the hotspot substitution (T523S) produced a receptor (*Dmel*Ir75b$^{DsecS1,T523S}$) that is more similar in response profile to *Dsec*Ir75b than either *Dmel*Ir75b$^{DsecS1}$ or *Dmel*Ir75b$^{T523S}$ (*Figure 5A*). However, this receptor still has robust sensitivity to C4, like *Dmel*Ir75b but unlike *Dsec*Ir75b (*Figure 5A*). Further incorporation of the three additional substitutions of residues in the S2 domain (*Prieto-Godino et al., 2017*), generated a receptor *Dmel*Ir75b$^{DsecS1,4mutS2}$ that, in terms of specificity, is indistinguishable from *Dsec*Ir75b (*Figure 5A*). However, this receptor has overall reduced sensitivity for all acids when compared with *Dsec*Ir75b, and its response to C6 is significantly smaller than that of *Dsec*Ir75b (*Figure 5A*). Despite this overall reduced sensitivity, *Dmel*Ir75b$^{DsecS1,4mutS2}$ still has a significantly increased sensitivity to C6 when compared with *Dmel*Ir75b (*Figure 5A*).

To visualize how the S1 residues might impact receptor function, we mapped the position of the six derived changes within this lobe onto the predicted *Dmel*Ir75b structure (*Figure 5B*). All of these are located on the surface of the LBD and therefore unlikely to contact ligands directly. Irs and iGluRs are thought to exhibit the same global structure and stoichiometry (*Abuin et al., 2011*; *Abuin et al., 2019*), prompting us to align the *Dmel*Ir75b LBD onto a homotetrameric iGluR structure (PDB 3KG2) (*Figure 5C*). This analysis did not suggest particularly close proximity of most of these residues to the interaction interface between subunits (*Figure 5C*).

## Discussion

As species adapt to new ecological niches, olfactory receptors evolve to define novel relationships between external signals and internal neural representations. Population genetic analyses of *D. melanogaster* strains isolated from diverse global habitats reveal that olfactory receptors (and other chemosensory protein families) display some of the strongest genomic signatures of recent selection (*Arguello et al., 2016*), suggesting these proteins act as 'first responders' in local adaptation to new environments. In this work, we have used comparative sequence and functional analyses across the well-defined phylogeny of the *Drosophila* genus to study how members of the organic acid-sensing clade of Irs have changed over evolutionary timescales.

Our most important finding is the discovery of a 'hotspot' residue whose mutation had a major effect on the odor specificity of one receptor (Ir75a) at two different timepoints during species

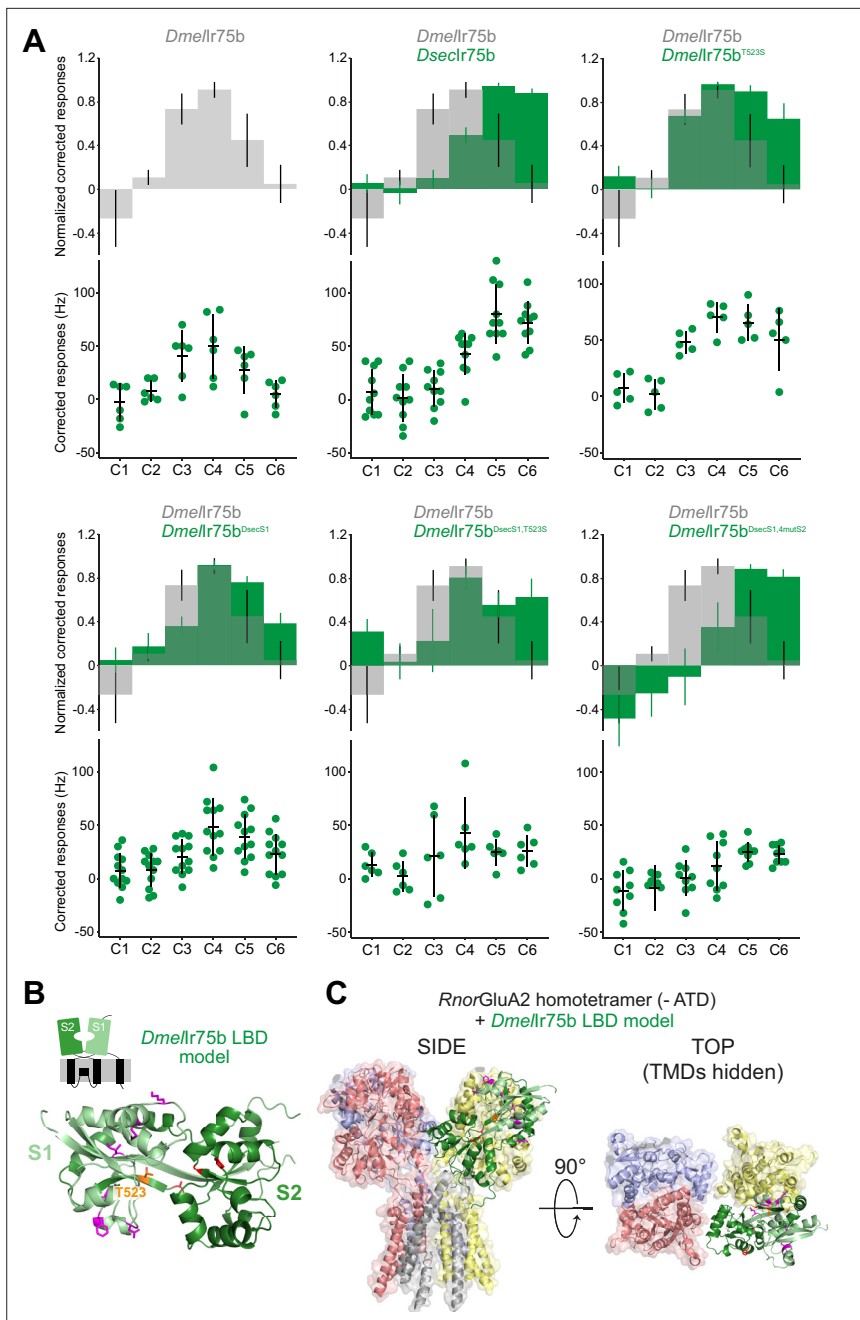

**Figure 5.** Epistasis in the ligand-binding domain of Ir75b. (**A**) Quantification of responses of the indicated wild-type and mutant Ir75b variants expressed in the Ir decoder neuron. The *Dmel*Ir75b[DsecS1] variant includes the six derived changes between *D. sechellia* and *D. simulans*/*D. melanogaster* as well as a seventh amino acid substitution common to *D. sechellia* and *D. simulans* receptors. (**B**) Protein model of the *Dmel*Ir75b ligand-binding domain (LBD) (as in **Figure 4C**). The S1 and S2 lobes are colored pale and dark green, respectively. Surface-located amino acids that differ between *Dmel*Ir75b and *Dsec*Ir75b in the S1 domain are depicted in magenta; the hotspot (position 523) is depicted in orange and other residues in the ligand-binding pocket in the S2 domain that are different in *Dsec*Ir75b are shown in red. (**C**) *Dmel*Ir75b LBD model (colored as in (**B**)) aligned to the full-length homotetrameric X-ray crystal structure of *Rnor*GluA2 (PDB accession 3KG2), which is colored by chain in faded gray, red, blue, and yellow (the LBD of the gray chain is hidden to facilitate visualization of the *Dmel*Ir75b LBD (green) in this position). The *Rnor*GluA2 amino-terminal domain, which is not present in most Irs (**Rytz et al., 2013**), has been removed for clarity.

The online version of this article includes the following figure supplement(s) for figure 5:

**Source data 1.** Source data for **Figure 5A**.

diversification, as well as on the tuning change of a distinct receptor (Ir75b). For Ir75a, our data support a model in which the ancestral drosophilid Ir75a was – contrary to previous assumptions – predominantly a C4 sensor that switched, through mutation at the hotspot (and other sites), to preferential detection of C2 in the *melanogaster/obscura* ancestor before 'reverting' to C4-sensing in *D. sechellia*. The ancestral function of Ir75b is still unclear: we have not been able to unambiguously identify Ir75b neurons across the drosophilid phylogeny, as they are not easily distinguishable from those expressing a related receptor, Ir75c (**Prieto-Godino et al., 2017**). However, like Ir75a, Ir75b has adapted through hotspot substitution along the lineage leading to *D. sechellia*. Neither the identity nor even the chemical class of amino acids occupying the hotspot are conserved in these receptors (phenylalanine [F, aromatic] or leucine [L, hydrophobic] in Ir75a; threonine [T] or serine [S; both polar] in Ir75b). These observations indicate that the position of the hotspot in these Irs' LBDs, and not its identity, must explain its central role in defining odor response properties in different receptors. Studies of other Irs and distinct families of olfactory receptors in invertebrates and vertebrates have revealed enormous inter- and intraspecific sequence variation, which has been linked in some cases to differences in odor tuning (**Adipietro et al., 2012**; **Auer et al., 2020**; **Block, 2018**; **Butterwick et al., 2018**; **Del Mármol et al., 2021**; **Leary et al., 2012**; **Mainland et al., 2014**; **Prieto-Godino et al., 2017**; **Yang et al., 2017**). It will be interesting to examine whether the hotspot position of Ir75a and Ir75b is relevant for understanding the evolution of ligand specificity of other Irs, and whether analogous hotspots exist in different receptor types, revealing favored (or constrained) mechanisms through which new odor detection properties evolve. Indeed, recent characterization of the functional divergence of Odorant receptor 22 a (Or22a) in *D. melanogaster* and *D. sechellia* identified a key position that contributes to determining tuning specificity in other insect Ors (**Auer et al., 2020**).

The mechanistic role of the hotspot is unclear, as we still know very little about how ligands are recognized by Irs. Ir75a orthologs across the drosophilid phylogeny conserve all three main agonist-binding residues characteristic of iGluRs (**Benton et al., 2009**; **Mayer, 2006**), suggesting that the core contacts between odors and this receptor are similar to the ancestral ligand-binding mechanism of this superfamily. One of these residues, an arginine (R297 in *Dmel*Ir75a), is globally conserved in diverse acid-sensing Irs, but not in nonacid-sensing receptors; this residue has previously been speculated to interact with the carboxyl group of acidic odors, analogous to the interaction of iGluRs and the α-carboxyl group of the glutamate ligand (**Figure 4C**; **Abuin et al., 2011**). Consistently, mutation of this residue in one receptor (Ir84a) completely abolishes ligand-evoked responses (**Abuin et al., 2011**). The hotspot in Ir75a and Ir75b is located at some distance from this arginine (predicted >9 Å, **Figure 4C**), which, together with the less drastic effect of its mutation, suggests the hotspot may only indirectly impact ligand–receptor interactions and/or influence ligand-induced LBD conformational changes. The latter mechanism may be a more subtle way of modifying the tuning profile of these receptors without the risk of complete loss of function. Determination of the mechanistic impact of hotspot mutations is an important future priority, but will likely require experimentally determined structures of odor-free and odor-bound Ir LBDs. Such knowledge may also inform our understanding of the mechanism of ligand-induced gating in iGluRs, where the equivalent position (Y732 [**Figure 4C**]) is thought to have a function in ligand-induced channel gating although its precise role is unclear (**Armstrong and Gouaux, 2000**; **Mamonova et al., 2008**).

While the hotspot is clearly important, its contribution to modification of tuning properties is shaped by additional changes in these receptors. For Ir75a, there are two additional substitutions within the ligand-binding pocket, while for Ir75b, one or more residues located on the external surface of the LBD are relevant. In both cases, the functional consequences of combining these substitutions with that of the hotspot are not easily predicted from their individual impact, revealing complex epistatic interactions. Moreover, the distance of these additional residues from the predicted odor-binding site suggests the existence of allosteric effects of certain amino acid positions on odor/receptor interactions. These results are of interest in light of molecular evolutionary analyses of olfactory receptor repertoires, which have identified numerous residues under positive selection (or relaxed purifying selection) in different receptors – implying a contribution to functional divergence – in regions far from the predicted ligand-binding pocket (**Arguello et al., 2016**; **Chen et al., 2010**; **Gardiner et al., 2009**; **Smadja et al., 2009**; **Steiger et al., 2010**). Together, such observations argue that the evolution of olfactory receptor specificity does not simply arise by alterations in direct contacts of receptors with

odor ligands, but rather can emerge from a complex network of interactions of amino acid substitutions – with major or minor effects – both near and far from the ligand-binding site.

One clear limitation of our study is the restriction of our profiling to a set of linear carboxylic acids at a single concentration. While these ligands are found in nature and are the best agonists for these receptors among many screened odors (*Silbering et al., 2011*), they necessarily only give a partial insight into the functional changes of individual receptors. The C4 to C6 switch of Ir75b in *D. sechellia* is likely related to high abundance of C6 in its sole host fruit (noni). For Ir75a, the ecological framework is less clear: we speculate that the C4 to C2 switch in the *melanogaster/obscura* ancestor may be related to the use of host fruits with acetic acid producing bacteria. Such sensitivity may have been no longer relevant for *D. sechellia* which preferentially feeds upon ripe (nonfermenting) noni fruit (which contains much more C4 than C2 [*Auer et al., 2020*; *Farine et al., 1996*]). Future expansion of the odor profiling of these species' receptors will be essential to understand the pressures in the natural world that have selected for olfactory receptor proteins with new chemical recognition properties. Such knowledge, together with mechanistic insight into ligand/receptor interactions, may be useful to reengineer the ligand-binding specificities of receptors to create, for example, chemogenetic tools (*Fukabori et al., 2020*) or design pharmacological manipulators to control the olfactory-guided behaviors of pest insects.

## Materials and methods
### *Drosophila* strains and culture
Flies were maintained on a standard culture medium at 25 °C in 12 hr light:12 hr dark conditions. We used the following published *D. melanogaster* strains: $Ir84a^{Gal4}$ (*Grosjean et al., 2011*), *UAS-DmelIr75a*, *UAS-DsecIr75a*, *UAS-DmelIr75a$^{T289S,Q536K,F538L}$* (*Prieto-Godino et al., 2016*). Other drosophilid species were obtained from the *Drosophila* Species Stock Center: *D. sechellia* (14021-0248.25), *D. simulans* (14021-0251.195), *D. yakuba* (14021-0261.01), *D. erecta* (14021-0224.01), *D. ananassae* (14024-0371.13), *D. pseudoobscura* (14011-0121.94), *D. willistoni* (14030-0811.24), *D. mojavensis* (15081-1352.22), and *D. virilis* (15010-1051.87).

### Molecular biology
cDNAs of *D. melanogaster* and *D. sechellia* Ir75a and *Ir75b* were previously described (*Prieto-Godino et al., 2017*; *Prieto-Godino et al., 2016*). Site-directed mutagenesis was performed using standard procedures, and mutant cDNAs were subcloned into *pUAST attB* for transgenesis of *D. melanogaster* using the phiC31 site-specific integration system (landing site attP40) by BestGene Inc and Genetic Services Inc. All transgenes were sequence verified both before and after transformation.

### Ancestral protein reconstruction
The 16 'modern' sequences of Ir75a shown in *Figure 2—figure supplement 1* were used to computationally infer the ancestral sequences at all of the nodes of the tree using the known phylogeny. Ancestral sequences were calculated using FastML (*Ashkenazy et al., 2012*) with rate variation modeled as a gamma distribution. The marginal posterior probability for most of the amino acids of all of the reconstructed nodes was above 0.8 (and the maximum, 1.0, for the hotspot position). To ascertain the robustness of the inferences made by FastML, we also inferred ancestral sequences of the nodes using codeml, within the PAML package. The inferred sequences from these independent analyses had a global identity of 96.5 % and were identical at the three amino acid positions under study. DNA sequences encoding the Ir75a$^{Dros}$ and the Ir75a$^{obs-mel}$ sequences predicted by FastML were synthesized by Eurofins-Genomics, subcloned into *pUAST attB* and transformed into flies as described above.

### Analysis of intraspecific variation in *Ir* genes
Polymorphisms in *Ir75a* were analyzed in published sequence datasets for *D. melanogaster* (848 strains; *Hervas et al., 2017*), *D. simulans* (90 strains; *Signor et al., 2018*), and *D. sechellia* (46 strains; *Schrider et al., 2018*), but no informative variation was detected within the codons encoding functional determinants of odor specificity in Ir75a defined in this work (five *D. melanogaster* strains had a synonymous SNP [TTT] in the TTC codon that encodes F538).

## In vivo electrophysiology

Single sensillum extracellular recordings were performed essentially as described in **Benton and Dahanukar, 2011**; **Prieto-Godino et al., 2017**. For all stimuli, 10 μl odor (1% vol/vol in solvent) were used, and presented to the animal in a 500 ms pulse. ac2 sensilla were identified in different species by targeting sensilla at antennal locations where ac2 are found in *D. melanogaster*, and using pyridine as a diagnostic odor, which activates the ac2 neuron expressing Ir41a and appears to be conserved across species (*Figure 1B*). CAS numbers of odors are as follows: formic acid (64-18-6), acetic acid (64-19-7), propionic acid (79-09-4), butyric acid (107-92-6), pentanoic acid (109-52-4), hexanoic acid (142-62-1), pyridine (110-86-1) (diagnostic for ac2), and octanol (111-87-5) (diagnostic for ac3). Odor-evoked responses were calculated by summing the activity of all OSNs in a sensillum to a given stimulus, as reliable spike sorting is not possible and other neurons housed in the ac2 or ac4 sensilla do not respond to acidic odors (*Silbering et al., 2011*). We counted the number of spikes in a 500 ms window at stimulus delivery (200 ms after stimulus onset due to a delay introduced by the air path), subtracted the number of spikes in a 500 ms window 2 s before stimulus delivery, and doubled the result to obtain spikes/s. To calculate solvent-corrected responses (as shown in the figures), we subtracted from the response to each diluted odor, the response obtained when stimulating with the corresponding solvent, water for all odors except for pyridine and octanol, which were dissolved in paraffin oil (8012-95-1). A maximum of four sensilla were tested per animal, and individual genotypes were measured, in an interleaved fashion, on multiple independent days. Normalized responses were calculated by dividing solvent-corrected responses of a given sensillum to each odor by the maximal response of that sensillum, such that each sensillum always had one odor whose normalized response was 1. A prerequisite for PCA analysis is to input *z*-scored responses; these were calculated with the in-built function of MATLAB, which works according to the definition of *z*-scoring, that is, across each sensillum recording, each response was subtracted from the mean response of all sensillar responses and divided by the standard deviation.

## Statistical analysis

PCA was carried with in-built MATLAB functions. Statistical analyses were carried out with in-built functions in R Studio or Igor. For all statistical tests, a Shapiro test for normality was first performed. If both samples being compared were normally distributed, a *t*-test was performed; if one of the samples was not normally distributed, a Wilcoxon test was run. When performing multiple comparisons, p values were corrected using the Bonferroni method. Epistasis was calculated by determining whether the effects of each individual mutation added linearly, by comparing the observed responses of the combination of two (or three) mutations with the expected distribution of responses if those mutations were added linearly. Briefly, to generate expected response distributions for the linear combination of the mutations, we subtracted the mean responses of the initial receptor from the two 'intermediate' receptors, which provides the 'effect' of each mutation(s) individually; we then we took 1000 random samples from the 'effects' of each of the two mutations and summed them to generate a distribution of the expected responses if the two effects of each mutation added linearly. To determine whether two mutations interacted epistatically, we statistically compared the expected distribution with the actual distribution (the double (or triple) mutant combination) using a Wilcoxon test. If the result of this test was p < 0.05 after correction for multiple comparisons using Bonferroni method, the two mutations were considered to interact epistatically.

## Protein model visualization and comparison

*Dmel*Ir75a and *Dmel*Ir75b models were predicted by AlphaFold (*Jumper et al., 2021*). The X-ray crystal structure of *Rnor*GluA2 bound to glutamate was PDB accession 1FTJ (*Armstrong and Gouaux, 2000*); the full-length *Rnor*GluA2 X-ray crystal structure was PDB 3KG2 (*Sobolevsky et al., 2009*). Structure visualization was performed using PyMol version 2.3.3. The *Dmel*Ir75b LBD was aligned to the *Rnor*GluA2 homotetramer using the 'align' command in PyMol.

## Acknowledgements

We thank Roman Arguello, Jean Hausser, Michael Shahandeh, and other members of the Benton laboratory for discussions and comments on the manuscript. We thank Tom Baden for help with Igor

analysis, Roman Arguello and Michele Marconcini for help with sequence polymorphism analysis, and Liliane Abuin and Steeve Cruchet for technical assistance. HRS was supported by a European Molecular Biology Organisation Long-Term Fellowship (ALTF 940-2019) and a Helen Hay Whitney Foundation Fellowship. LLP-G's laboratory was supported by a European Research Council (ERC) Starting Investigator Grant (802531), and the Francis Crick Institute, which receives its core funding from Cancer Research UK (FC001594), the UK Medical Research Council (FC001594), and the Wellcome Trust (FC001594). Work in RB's laboratory was supported by ERC Consolidator and Advanced Grants (615094 and 833548, respectively), a Human Frontier Science Program Young Investigator Award (RGY0073/2011), and the Swiss National Science Foundation Nano-Tera Envirobot project (20NA21_143082).

## Additional information

### Funding

| Funder | Grant reference number | Author |
|---|---|---|
| EMBO | ALTF 940-2019 | Hayden R Schmidt |
| Helen Hay Whitney Foundation | | Hayden R Schmidt |
| European Research Council | 802531 | Lucia L Prieto-Godino |
| Cancer Research UK | FC001594 | Lucia L Prieto-Godino |
| Medical Research Council | FC001594 | Lucia L Prieto-Godino |
| Wellcome Trust | FC001594 | Lucia L Prieto-Godino |
| European Research Council | 615094 | Richard Benton |
| European Research Council | 833548 | Richard Benton |
| Swiss National Science Foundation | 20NA21_143082 | Richard Benton |
| Human Frontier Science Program | RGY0073/2011 | Richard Benton |

The funders had no role in study design, data collection and interpretation, or the decision to submit the work for publication.

### Author contributions

Lucia L Prieto-Godino, Conceptualization, Data curation, Formal analysis, Funding acquisition, Investigation, L.L.P.-G. designed and performed all experiments and analyses, except for the protein visualization., Methodology, Project administration, Visualization, Writing - original draft, Writing - review and editing; Hayden R Schmidt, Funding acquisition, H.R.S. performed the protein visualization analysis., Investigation, Visualization, Writing - review and editing; Richard Benton, Conceptualization, Funding acquisition, Project administration, Resources, Supervision, Writing - original draft, Writing - review and editing

### Author ORCIDs

Lucia L Prieto-Godino ⓘ http://orcid.org/0000-0002-2980-362X
Richard Benton ⓘ http://orcid.org/0000-0003-4305-8301

### Decision letter and Author response

Decision letter https://doi.org/10.7554/eLife.69732.sa1
Author response https://doi.org/10.7554/eLife.69732.sa2

# Additional files

## Supplementary files
- Supplementary file 1. Nucleotide sequences encoding the reconstructed ancestral Ir75a orthologs.
- Transparent reporting form

## Data availability
All data generated or analyzed during this study are included in the manuscript and supporting files. Source data files have been provided for Figures 1, 2, 3 and 5.

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
