## [Editor Report]

This study investigates evolutionary changes in ligand preference that occur in an olfactory receptor (IR75a) across the *Drosophila* phylogeny. The authors find that IR75a displays different odor preferences, for acetic acid or butyric acid, across *Drosophila* species, and link odor preference to particular protein mutations in the receptor. Reconstruction of a putative ancestral IR75a revises the timeline for IR75a evolution, and structural modeling suggests how mutations alter odor preference.

---

## [Decision Letter]

**Decision letter after peer review:**

Thank you for submitting your article "Molecular reconstruction of recurrent evolutionary switching in olfactory receptor specificity" for consideration by *eLife*. Your article has been reviewed by 3 peer reviewers, one of whom is a member of our Board of Reviewing Editors, and the evaluation has been overseen by Catherine Dulac as the Senior Editor. The reviewers have opted to remain anonymous.

Essential revisions:

As you will see below, all reviewers were enthusiastic about the study, but noted that certain aspects of the paper required additional attention. In preparing your revision, please prioritize the following items. I have also included all reviewer comments for your benefit.

1) All reviewers had concerns about the structural modeling and ligand docking, and suggested either substantial additional validation or removal of this portion of the paper.

2) All reviewers agreed that analysis of ligand preference based on a single concentration was insufficient, and asked for more detailed receptor/ligand dose response curves.

*Reviewer #1 (Recommendations for the authors):*

1. Conclusions about odor preference of the ancestral receptor presumably boil down to whether position 538 in the ancestral receptor is F or L. Does mutation of that one position flip the ligand preference of the ancestral receptor, and if so, what is the level of mathematical confidence for assignment of L rather than F at this position in the ancestral state?

2. IR75a assignments in Figure 1 appear to be based on anatomical proximity to pyridine-responsive sensilla. It seems possible that this anatomical arrangement could change in divergent species. The authors should provide additional supportive evidence that C4 preference in the measured sensilla of D. Wil and D. moj is in fact due to IR75a; one way to do this would be to test in the empty neuron model used in Figure 2.

3. The authors analyze responses at a single concentration- the sensitivity and amplitude of olfactory receptor responses amplitude are not always correlated across ligands. It would be useful to extend analysis in Figure 2 and/or Figure 1 to a few additional test concentrations.

4. Two aspects of the manuscript: the structural modeling and evolutionary timeline (p5-6) are pretty speculative. The authors appropriately acknowledge the caveats of modeling in the discussion, yet conclusions based on these models may still be highlighted too much. The evolutionary timeline seems like even more of a reach as it is not yet clear whether changes in ligand preference are capable of driving selective pressure on evolution of these receptors (in the absence of behavioral experiments), or whether ligands analyzed are the most ecologically relevant.

*Reviewer #2 (Recommendations for the authors):*

Following up on the 3 points raised earlier, it would be most insightful to see whether intraspecific variation matches their hypotheses on the fixation rate or frequency of the analyzed polymorphisms. This would provide an independent source of validation of their analysis, as well as ecological support to the premise of focusing on C2-C4 ligands, addressing most of the points raised earlier as weaknesses. Datasets that contain population genomic data are available for *D. melanogaster* and *D.sechellia* (generated by the Benton group and others), and the authors mention briefly that they were unable to retrieve relevant information, I wonder if they could expand on their findings when parsing these data.

In terms of docking results, I am inclined to suggest removing this section altogether, as I truly believe the remainder of the work is extremely valuable on its own. However, if the authors choose to keep it, I think this section would be stronger if there was more extensive characterization to validate the proposed models. For example, the authors dock various C2 and C4 ligands onto a Dmel homology model of Ir75a. Since these experiments are in silico and relatively fast and inexpensive, it would be helpful to dock other known ligands and evaluate whether an overall qualitative agreement between docking scores and ligand activity can be found. Alternatively, they could generate models of the orthologous Ir75a proteins in the clade for which they have collected experimental data, and dock C2-C4 ligands and attempt to find agreement between modeling results and experimental results.

An alternative approach that would increase the value of this section would be to show that indeed the identity of the 538 residue impacts perhaps the neuronal behavior in terms of spiking dynamics or amplitudes, or otherwise any other aspect that could show that the coupling between ligand-binding and receptor function differs between mutants. This is I imagine less trivial than extending the modeling attempts, so it is entirely optional, but is an alternative that would help support the idea that this residue does not immediately interact with the ligand but rather has a coupling role.

*Reviewer #3 (Recommendations for the authors):*

1. The definition of 'hotspot' is unclear. As the authors mentioned in Discussion, if they want to say that this is a 'hotspot' residue in the IR family and to expand the argument to the entire IR family, other IRs should be examined. If they want to argue in terms of evolution whether it is random or positive under the selective pressure, evidence is not enough. Although the biological meaning of IR75a in Dsec is unclear, the authors seem to be claiming that C4-specific evolution is a positive selection. These points regarding 'hotspot' should be carefully discussed in the paper.

2. In the same line with the above comment, the authors should give discussion about the recent report that tuning of Or22a is important for attraction to noni and thus, a molecular evolution of Or22a appears to affect food preference and selection (Nature 579, 402-408, 2020).

3. In Figure 1c, the response amplitude in Dmel and Dsec to C2 looks similar, and thus, I wonder whether we can conclude that Dmel but not Dsec is 'C2-sensing'. Since the responsiveness of Dmel IR75a is much weaker than other species, it may be a unique receptor and have evolved independently.

4. Why is the responsiveness of Dsec to C4 in Figure 1C and 2C different (i.e. 200 Hz in Figure 1C and 70 Hz in Figure 2C)? Is it possible that the complex with endogenous co-receptor DmelIR8a is not properly formed? I wonder whether the responsiveness in *Drosophila* other than Dmel is indeed recapitulated in 'IR decoder neuron'.

5. A289 also appears in the course of C4-C2 evolution, and further, this residue seems to be involved in ligand binding according to the double mutation result in Figure 3. Therefore, I am curious about T289A.

6. In Docking simulation (Figure 4), only the binding mode of the top rank is argued. Since the homology modeling is not accurate, at least top 10 stable models should be analyzed.

7. In Figure 4C, the docking score is better for C4 than C2, which is inconsistent with the responsiveness of DmelIR75a.

8. In comparison with the GluA2 LBD/glutamate co-crystal, the authors mention 'equivalent to that of the a-carboxyl group of the glutamate ligand with a conserved arginine in iGluRs (line 313-314), but it looks to me that the angle of the binding is quite different,. Based on this observation, I am not sure whether it can be concluded that F538 and T523 are not involved in ligand binding.

9. I would like to see the docking simulation of F528L to see whether the simulation method itself is reliable.

10. Docking simulation for IR75b would be a good comparison, since the ligand selectivity is similar with IR75a.

11. I cannot find a logic for choosing amino acids for mutation in Figure 5A. In addition, we cannot tell which amino acid is important for ligand selectivity from the swapping of the entire S1.

12. The X-axis label in Figure 1e is wrong?

13. The color is hard to be distinguished in Figure 3B and C.

14. In Figure 5B, DsecIR75b should be read DmelIR75b?

---

## [Author Response]

Essential revisions:As you will see below, all reviewers were enthusiastic about the study, but noted that certain aspects of the paper required additional attention. In preparing your revision, please prioritize the following items. I have also included all reviewer comments for your benefit.1) All reviewers had concerns about the structural modeling and ligand docking, and suggested either substantial additional validation or removal of this portion of the paper.

We fully acknowledge these concerns; our goal in the initial submission was to provide simple “visuals” of the position of the hotspot in the LBD and the potential location of bound odor ligands. Although we tried to be very cautious about our interpretation of these analyses, we agree that the evidence supporting our claims was weak.

In this revision, we have replaced the protein models with new models generated using the AlphaFold algorithm. Although the lack of any experimentally-determined structures of these Irs precludes definitive comparison of model quality, the ab initio AlphaFold protein structure predictions appear to be superior to our previous protein homology models (Author response image 1) . For example, secondary structural elements have better continuity (in the previous models, there were several secondary structural elements that were connected by “spaghetti-like” strings of residues), and the amino-acid side chains generally have more natural poses (in the previous models, certain side chains had odd geometries or were buried unnaturally in pockets). Thus, we consider presenting the new models to show the relative location of various residues within the context of the LBD and the putative subunit interfaces is still useful.

**Author response image 1. sa2fig1:** 

We repeated the ligand docking analyses using the AlphaFold models. While ligands could be docked deeper in the binding pocket – which might be consistent with more plausible poses – and often showed the interaction between the carboxyl group with the conserved Arg residue (as highlighted in the first submission), several poses revealed less clear physical separation of ligand and hotspot. As in our original analysis, the docking scores of different poses was rather similar, preventing any sort of confident ranking. Beyond the caveats of ligand docking into protein models, the docking software was designed to best dock ligands with low μM or even nM affinity, while limited pharmacological data on other Irs indicate that these receptors can have EC50 values of several 10s-100s μM (e.g., PMID 28067294). As such, we agree with the reviewers’ caution regarding the docking, and do not include any such analyses in the revised manuscript. All speculation about the mechanistic role of the hotspot is limited, briefly, to the Discussion.

2) All reviewers agreed that analysis of ligand preference based on a single concentration was insufficient, and asked for more detailed receptor/ligand dose response curves.

We fully agree, as we acknowledge in the Discussion, that the single concentration used for the experiments only provides a partial view of the changes in odor tuning profile.

We have now generated dose-response curves for endogenous ac2 Ir75a neurons in *D. melanogaster* and *D. sechellia* (the two key comparison species of the work) for acetic acid (C2), propionic acid (C3) and butyric acid (C4) (Author response image 2) .

Several observations can be made from these data. First, the dynamic range of neuronal responses is relatively narrow: for example, for *D. melanogaster* ac2, none of these ligands evoke clear responses at the 10^-4^ (v/v) concentration. The low sensitivity of Ir neurons to their ligands (compared, for example, to Or neurons) has previously been recognized (e.g., PMID 23162431, Figure 1). Second, we find the highest odor doses (10^-1^) evokes, for several neuron/odor combinations, a smaller number of spikes than lower odor concentrations (e.g., C3 and C4 in *D. sechellia*). We do not think this non-linearity reflects an unusual pharmacological property of the Ir itself, but rather technical limitations of in vivo recordings from insect olfactory sensilla. High odor doses can lead to changes in spike amplitude (either “pinching” or increases) and non-specific activation of multiple neurons within a sensilla (with the potential for ephaptic inhibition between these neurons, e.g., PMID 23172146). Determination of the isolated spiking response of a given neuron to an odor can therefore be difficult; we also note that spiking is a fairly distal signaling event from the biochemical process of odor activation of the receptor. As such, defining the real “saturated” neuronal responses – to be able to calculate neuronal/receptor pharmacological properties, such as EC50 – is not trivial in vivo, at least in Ir neurons. Despite these drawbacks, we highlight that relative potency of the ligands is maintained across the concentrations tested, e.g., C2 always evokes the highest responses in *D. melanogaster* Ir75a neurons, and C4 always evokes the highest response in *D. sechellia* Ir75a neurons (and vice versa). Thus, we believe our analysis of a single concentration still captures the fundamental differences in tuning profile of these receptors. We have made substantial efforts to pharmacologically characterize these Irs in *Xenopus oocytes*, through two-electrode voltage clamp recordings, with the goal of determining the contribution of specific residues to ligand/receptor affinity and/or channel gating efficiency. Unfortunately, difficulties to obtain reliable Ir-dependent current responses to odor presentation, and to establish the plateau of dose response curves – as was already evident from our first published TEVC recordings (PMID 21220098, Figure 4F) preclude clear conclusions. Similar efforts to functionally express in Irs in cultured mammalian cells has proven even less successful. Regarding the choice of concentration, the starting point of our work was in *D. melanogaster*, where the 10^-2^ concentration effectively separates the responses of C2, C3 and C4, while avoiding the potential issues with higher stimulus concentrations. 10^-2^ is a very standard concentration used in the field, though of course we note that the (unknown) number of odor molecules reaching the antennae also depends upon the method of stimulus delivery (e.g., odor volume, air flow). Whether the 10^-2^ stimulus is ecologically relevant is difficult to know, but we do point out that as this concentration evokes a sub-maximal neuron spiking frequency, it is presumably within the range of what a neuron can respond to in nature. Moreover, natural odor sources can have high chemical concentrations, such as vinegar (where the proportion of C2 can be 4-8% or more) and noni fruit (where hexanoic acid (C6), may represent up to 40% of the volatiles, PMID 32132713). We do acknowledge that, based upon the data in Author response image 1, for *D. sechellia* Ir75a neurons, 10^-3^ better discriminates responses to different odors. However, the vast majority of analyses in our paper use the Ir decoder system, where spiking responses of Ir75a receptor variants are generally lower, so 10^-2^ is likely to have been a good compromise.We hope these data and discussion will satisfy the reviewers. We fully acknowledge that, short of repeating all the experiments in this manuscript with dose-response curves of all receptors to all odors (which may or may not produce biologically meaningful “curves” due to the issues described above), we have only a partial view of the functional properties of these receptors. Nevertheless, we do not think that this limitation impacts the most important conclusions of our work.

Reviewer #1 (Recommendations for the authors):1. Conclusions about odor preference of the ancestral receptor presumably boil down to whether position 538 in the ancestral receptor is F or L. Does mutation of that one position flip the ligand preference of the ancestral receptor, and if so, what is the level of mathematical confidence for assignment of L rather than F at this position in the ancestral state?

We do not know whether mutation of this one position in the ancestral receptor is sufficient to flip the specificity, though we agree this would be a simple prediction we can test in the future. However, in all reconstructions L538 has a marginal posterior probability of 1 (i.e., maximal confidence).

2. IR75a assignments in Figure 1 appear to be based on anatomical proximity to pyridine-responsive sensilla. It seems possible that this anatomical arrangement could change in divergent species. The authors should provide additional supportive evidence that C4 preference in the measured sensilla of D. Wil and D. moj is in fact due to IR75a; one way to do this would be to test in the empty neuron model used in Figure 2.

The Ir75a neuron assignments are based upon (i) presence in the morphologically-distinguishable coeloconic sensilla, (ii) stereotyped spatial location of the ac2 sensillum class on the antennal surface, and (iii) pairing of these acid-sensing neurons with a pyridine-responsive neuron (diagnostic of Ir41a neurons). Changes in neuron pairing in the drosophilid genus appears to be incredibly rare, as indicated by surveys of multiple sensilla types in multiple species (e.g., PMID 28111079 and 14667348). In fact, we do not know of *any* cases where the grouping of neurons within a sensillum has changed between these species. This may be because such an adaptation would require developmental reprogramming of these lineages, which is genetically presumably more complex than “simple” re-tuning of a receptor. Of particular relevance to our study, a previous survey in *D. mojavensis* of responses of Ir-expressing coeloconic sensilla OSNs to a large odor panel revealed global strong conservation compared to *D. melanogaster* (PMID 30107159). (We note that in this study acetic acid was not included in the odor panel, so the important switch of odor specificity that we describe in our work was not appreciated). We believe it is highly unlikely therefore that there has been a switch in receptor identity.

We have attempted to visualize the pairing of Ir75a and Ir41a neurons in ac2, but RNA FISH for Ir41a has never been successful (perhaps due to low expression levels). We acknowledge that definitive evidence that the responses we assign to the “Ir75a neuron” in the different species is due to Ir75a function would require mutation of these genes in these species, or misexpression of the receptor genes in *D. melanogaster*, but we do not currently have the necessary genetic reagents.

3. The authors analyze responses at a single concentration – the sensitivity and amplitude of olfactory receptor responses amplitude are not always correlated across ligands. It would be useful to extend analysis in Figure 2 and/or Figure 1 to a few additional test concentrations.

We respond to this point in the general comments above. Importantly, the relative sensitivity to different odors is maintained across the range of concentrations tested.

4. Two aspects of the manuscript: the structural modeling and evolutionary timeline (p5-6) are pretty speculative. The authors appropriately acknowledge the caveats of modeling in the discussion, yet conclusions based on these models may still be highlighted too much.

As described above in the general comments, we now present what we consider are better protein models to illustrate the relative position of different residues in the LBD but have removed all docking analyses.

The evolutionary timeline seems like even more of a reach as it is not yet clear whether changes in ligand preference are capable of driving selective pressure on evolution of these receptors (in the absence of behavioral experiments).

We are not quite sure what the reviewer means by “whether changes in ligand preference are capable of driving selective pressure on evolution of these receptors” but if we interpret the sense of this statement correctly we agree we cannot (and have not) made any direct inference about how changes in ligand specificity may have occurred in response to environmental selection pressures through changes in reproductive success (which ultimately is what leads to alteration in allele frequencies in a population). Evidence for selection on specific residues can only be derived from population genetic studies, which is not possible for these sites (as they are unvarying within species in at least the sequenced populations of drosophilids). The “evolutionary timeline” we discuss concerns the potential order of mutations in Ir75a based solely upon their phenotypic contribution to the receptor response properties. To our knowledge, such questions have not been widely discussed in the olfactory receptor literature, though they are extensively studied in other types of evolving proteins. We were simply keen to air such interesting questions in this study system, while aiming to be appropriately circumspect about the strength of the evidence for any particular model.

Or whether ligands analyzed are the most ecologically relevant.

Whether the “best”/ecologically-relevant ligand has been identified is an issue at the crux of many olfactory studies. For the Irs we study here, our starting point was information from electrophysiological deorphanization screens of their neurons using diverse chemical classes of odors (PMID 21940430 and 16162917). The ligands we use evoke the highest (or among the highest) responses for each receptor; these chemicals are also found in relevant ecological contexts of the drosophilid species. We cannot of course exclude that better ones exist, but we presume that these are likely to be different types of carboxylic acids.

Reviewer #2 (Recommendations for the authors):Following up on the 3 points raised earlier, it would be most insightful to see whether intraspecific variation matches their hypotheses on the fixation rate or frequency of the analyzed polymorphisms. This would provide an independent source of validation of their analysis, as well as ecological support to the premise of focusing on C2-C4 ligands, addressing most of the points raised earlier as weaknesses. Datasets that contain population genomic data are available for *D. melanogaster* and *D.sechellia* (generated by the Benton group and others), and the authors mention briefly that they were unable to retrieve relevant information, I wonder if they could expand on their findings when parsing these data.

We have now added additional information to the manuscript on the results of our (ultimately uninformative) survey of intraspecific genetic variation in these drosophilids.

In terms of docking results, I am inclined to suggest removing this section altogether, as I truly believe the remainder of the work is extremely valuable on its own. However, if the authors choose to keep it, I think this section would be stronger if there was more extensive characterization to validate the proposed models. For example, the authors dock various C2 and C4 ligands onto a Dmel homology model of Ir75a. Since these experiments are in silico and relatively fast and inexpensive, it would be helpful to dock other known ligands and evaluate whether an overall qualitative agreement between docking scores and ligand activity can be found. Alternatively, they could generate models of the orthologous Ir75a proteins in the clade for which they have collected experimental data, and dock C2-C4 ligands and attempt to find agreement between modeling results and experimental results.

As described above in the general comments, we now present what we consider are better protein models to highlight the relative position of different residues in the LBD but have removed all docking analyses. Neither previous, nor more recent, docking attempts, provided any clear insight into what determines the specificity of different receptor orthologs for one ligand compared to another.

An alternative approach that would increase the value of this section would be to show that indeed the identity of the 538 residue impacts perhaps the neuronal behavior in terms of spiking dynamics or amplitudes, or otherwise any other aspect that could show that the coupling between ligand-binding and receptor function differs between mutants. This is I imagine less trivial than extending the modeling attempts, so it is entirely optional, but is an alternative that would help support the idea that this residue does not immediately interact with the ligand but rather has a coupling role.

We are also very keen to gain mechanistic insight into the effect of these mutations, but such efforts are indeed not trivial and extend beyond the work of the current manuscript.

Reviewer #3 (Recommendations for the authors):1. The definition of 'hotspot' is unclear. As the authors mentioned in Discussion, if they want to say that this is a 'hotspot' residue in the IR family and to expand the argument to the entire IR family, other IRs should be examined. If they want to argue in terms of evolution whether it is random or positive under the selective pressure, evidence is not enough. Although the biological meaning of IR75a in Dsec is unclear, the authors seem to be claiming that C4-specific evolution is a positive selection. These points regarding 'hotspot' should be carefully discussed in the paper.

We have rephrased the Discussion text to clarify that our definition of hotspot pertains *only* to the Irs studied in our work. As there are many sequence differences between more divergent Irs, it is not easy to predict whether this site might be important in other receptors, but we leave this possibility open: “It will be interesting to examine whether the hotspot position of Ir75a and Ir75b is relevant for understanding the evolution of ligand-specificity of other Irs, …”.

2. In the same line with the above comment, the authors should give discussion about the recent report that tuning of Or22a is important for attraction to noni and thus, a molecular evolution of Or22a appears to affect food preference and selection (Nature 579, 402-408, 2020).

We now refer to this study in the same paragraph: “Indeed, recent characterization of the functional divergence of Odorant receptor 22a (Or22a) in

*D. melanogaster* and *D. sechellia* identified a key position that contributes to determining tuning specificity in other insect Ors (Auer et al., 2020).”.

3. In Figure 1c, the response amplitude in Dmel and Dsec to C2 looks similar, and thus, I wonder whether we can conclude that Dmel but not Dsec is 'C2-sensing'. Since the responsiveness of Dmel IR75a is much weaker than other species, it may be a unique receptor and have evolved independently.

The reviewer is correct that the *D. sechellia* ac2 neuron responses to C2 are comparable to those of *D. melanogaster* ac2, even though they are very much weaker than responses in *D. sechellia* to C4. Side-by-side comparison of endogenous neuronal responses in different species can be difficult, as other factors can impact absolute spiking rates (as the reviewer notes, for this wild-type strain of *D. melanogaster*, maximal spike frequencies are overall lower than for other species). Comparisons of responses of the receptor heterologously expressed in the Ir decoder neuron (Figure 2), more clearly shows that *Dmel*Ir75a responds more strongly to C2 than *Dsec*Ir75a. We have taken care in this revision to avoid the simplistic terminology that *Dmel*Ir75a is a “C2 sensor” and *Dsec*Ir75a is a “C4 sensor”, as it is clear from all the profiles that these receptors respond predominantly to one ligand but often more weakly to neighboring odors in the linear series of acids.

4. Why is the responsiveness of Dsec to C4 in Figure 1C and 2C different (i.e. 200 Hz in Figure 1C and 70 Hz in Figure 2C)? Is it possible that the complex with endogenous co-receptor DmelIR8a is not properly formed? I wonder whether the responsiveness in *Drosophila* other than Dmel is indeed recapitulated in 'IR decoder neuron'.

This difference in responsiveness is most easily explained by the fact that these recordings are from different neurons in different sensilla types (endogenous ac2 neuron vs heterologous ac4 neurons) as absolute number of spikes may be impacted by receptor protein expression levels, sensilla ultrastructure etc. The reviewer’s proposal that complex formation between *Dmel*Ir8a and an Ir from another species might be a further factor, but we suspect this has a fairly minor influence as Ir8a is highly conserved between *Dmel* and *Dsec* (97% amino acid identity); moreover, differential responses have been observed for *Dmel* receptors when expressed in their native and heterologous (but still *Dmel*) neuronal environment, e.g., Ir84a (PMID 21220098) and Ir75a (PMID 27776356).

5. A289 also appears in the course of C4-C2 evolution, and further, this residue seems to be involved in ligand binding according to the double mutation result in Figure 3. Therefore, I am curious about T289A.

We have not tested such a mutation, but it would certainly be interesting in the future to examine its functional significance for the more distant drosophilid species where A289 has appeared.

6. In Docking simulation (Figure 4), only the binding mode of the top rank is argued. Since the homology modeling is not accurate, at least top 10 stable models should be analyzed.7. In Figure 4C, the docking score is better for C4 than C2, which is inconsistent with the responsiveness of DmelIR75a.8. In comparison with the GluA2 LBD/glutamate co-crystal, the authors mention 'equivalent to that of the a-carboxyl group of the glutamate ligand with a conserved arginine in iGluRs (line 313-314), but it looks to me that the angle of the binding is quite different,. Based on this observation, I am not sure whether it can be concluded that F538 and T523 are not involved in ligand binding.9. I would like to see the docking simulation of F528L to see whether the simulation method itself is reliable.10. Docking simulation for IR75b would be a good comparison, since the ligand selectivity is similar with IR75a.

(Response to comments 6-10): As described above in the general comments, we now present what we consider are better protein models to indicate the relative position of different residues in the LBD but have removed all docking analyses.

11. I cannot find a logic for choosing amino acids for mutation in Figure 5A. In addition, we cannot tell which amino acid is important for ligand selectivity from the swapping of the entire S1.

The T523S (top right of this panel) is the “hotspot” mutation. In previous work (PMID 28111079) we showed that there were three additional derived changes near the pocket of the LBD (in the S2 domain) but when combined with the hotspot mutation, changes of these residues did not impact odor tuning, leading us to focus in the current work on the additional changes on the external surface of the LBD, which are all within the S1 domain (these had not been characterized previously). The set of mutations along the bottom row of panel A represent a test of these S1 residues alone (“DsecS1”) alone, in combination with the hotspot (“DsecS1,T523S”) and finally in combination with all of the LBD pocket residues in S2 (“DsecS1,4mutS2”).

The reviewer is correct that our data do not reveal which amino acid(s) are important within S1. The main concept we wish to emphasize is that a site or sites *outside* the binding pocket have important effects on receptor tuning properties. As the location of any of these residues on the surface of S1 residues is not obviously predictive of a function, we did not pursue further structure-function analysis in the current work, although this is certainly something that can be addressed in the future.

12. The X-axis label in Figure 1e is wrong?

This has been corrected.

13. The color is hard to be distinguished in Figure 3B and C.

We have adjusted the colors to increase the contrast between the overlaid blue and grey bars. We have also changed the coloration of bars in Figure 5A from blue to green to distinguish more clearly that these experiments display functional properties of variant versions of Ir75b.

14. In Figure 5B, DsecIR75b should be read DmelIR75b?

This model has now been replaced with the *Dmel*Ir75b LBD model generated using AlphaFold.